# A Comprehensive Investigation of the Structural, Thermal, and Biological Properties of Fully Randomized Biomedical Polyesters Synthesized with a Nontoxic Bismuth(III) Catalyst

**DOI:** 10.3390/molecules27031139

**Published:** 2022-02-08

**Authors:** Izabela M. Domańska, Anna Zgadzaj, Sebastian Kowalczyk, Aldona Zalewska, Ewa Oledzka, Krystyna Cieśla, Andrzej Plichta, Marcin Sobczak

**Affiliations:** 1Department of Biomaterials Chemistry, Chair of Analytical Chemistry and Biomaterials, Faculty of Pharmacy, Medical University of Warsaw, 1 Banacha Str., 02-097 Warsaw, Poland; izabela.domanska@wum.edu.pl (I.M.D.); ewa.oledzka@wum.edu.pl (E.O.); 2Department of Environmental Health Sciences, Faculty of Pharmacy, Medical University of Warsaw, 1 Banacha Str., 02-097 Warsaw, Poland; anna.zgadzaj@wum.edu.pl; 3Faculty of Chemistry, Warsaw University of Technology, 3 Noakowskiego Str., 00-664 Warsaw, Poland; skowalczyk@ch.pw.edu.pl (S.K.); aldona.zalewska@pw.edu.pl (A.Z.); andrzej.plichta@pw.edu.pl (A.P.); 4Institute of Nuclear Chemistry and Technology, 16 Dorodna Str., 03-195 Warsaw, Poland; k.ciesla@ichtj.waw.pl; 5Military Institute of Hygiene and Epidemiology, 4 Kozielska Str., 01-163 Warsaw, Poland

**Keywords:** biodegradable polymers, aliphatic polyesters, poly(ε-caprolactone), poly(l-lactide), poly(ε-caprolactone-*co*-glycolide), poly(l-lactide-co-ε-caprolactone), bismuth(III) 2-ethylhexanoate, ring opening polymerization

## Abstract

Aliphatic polyesters are the most common type of biodegradable synthetic polymer used in many pharmaceutical applications nowadays. This report describes the ring-opening polymerization (ROP) of l-lactide (L-LA), ε-caprolactone (CL) and glycolide (Gly) in the presence of a simple, inexpensive and convenient PEG200-BiOct_3_ catalytic system. The chemical structures of the obtained copolymers were characterized by ^1^H- or ^13^C-NMR. GPC was used to estimate the average molecular weight of the resulting polyesters, whereas TGA and DSC were employed to determine the thermal properties of polymeric products. The effects of temperature, reaction time, and catalyst content on the polymerization process were investigated. Importantly, the obtained polyesters were not cyto- or genotoxic, which is significant in terms of the potential for medical applications (e.g., for drug delivery systems). As a result of transesterification, the copolymers obtained had a random distribution of comonomer units along the polymer chain. The thermal analysis indicated an amorphous nature of poly(l-lactide-*co*-ε-caprolactone) (PLACL) and a low degree of crystallinity of poly(ε-caprolactone-*co*-glycolide) (PCLGA, *X*_c_ = 15.1%), in accordance with the microstructures with random distributions and short sequences of comonomer units (*l* = 1.02–2.82). Significant differences in reactivity were observed among comonomers, confirming preferential ring opening of L-LA during the copolymerization process.

## 1. Introduction

Bio-based polymeric materials are widely used in medicine and pharmacy (e.g., tissue engineering, drug delivery systems (DDSs), etc.). The most desirable are drug carriers derived from biocompatible polyesters, of which homo- and copolymers containing ε-oxycaproyl (Cap), glycolidyl (GG) and lactidyl (LL) units are the most commonly used biomaterials [1,2]. Concomitantly, the advantage of polyester drug carriers is their biodegradability. These polymers, once introduced into the organism, are well tolerated, metabolically decomposed, and eliminated via normal metabolic pathways [1]. The most attractive catalytic systems for the synthesis of polyesters are those consisting of metals (Zn, Sn, Zr, Fe). Nevertheless, the resulted polymers may contain traces of metal pollutants, leading to their high toxicity, which is undesirable for biomedical application [3]. As a result, they have been studied extensively over the last few decades, and significant progress has been made in terms of their synthesis using well-defined metalloorganic catalysts [4]. Among these, the use of alternative green catalysts has received particular consideration. Ring-opening polymerization (ROP) of lactones, such as L-lactide (L-LA), *rac*-lactide (*rac*-LA), ε-caprolactone (CL), and glycolide (Gly), using tin compounds as catalysts (e.g., tin(II) 2-ethylhexanoate (SnOct_2_)), is one of the methods of producing polyesters applied in medicine and pharmacy. Although the Food and Drug Administration (FDA) has approved SnOct_2_ as a food additive, tin(II) and tin(IV) ions or compounds tend to bind to the SH groups of proteins. In view of this, the catalyst is cytotoxic to some extent [5], and, thus, the resulting polymers should not be considered fully biocompatible [1]. Among other catalysts considered as non-toxic are Zn- and Zr- based compounds [3]. Our previous studies have shown that the polymers synthesized in the presence of diethylzinc [3] and zirconium(IV) acetylacetonate [6] may be considered as non-toxic in terms of cyto- and genotoxicity and thereby be suitable for pharmaceutical and medical applications.

Given the foregoing, in this work we will concentrate on bismuth(III) 2-ethylhexanoate (bismuth octoate, BiOct_3_), a potential nontoxic organometallic catalyst. Bi(III) is one of the ultratrace elements and its salts have long been used in medicine as both externally and internally administered drugs [4,7,8]. For example, bismuth(III) subsalicylate (BiSS) is a commercial drug for travelers’ diarrhea, nonulcer dyspepsia, and gastrointestinal complaints [7]. Furthermore, toxicity studies reveal that Bi(III) is not toxic even at the highest dose tested and it proved less toxic than Zn on cultured human kidney tubular cells [9]. According to Kowalik et al., bismuth-based complexes exhibit not only antimicrobial and anticancer activity, but recent results also reveal the ability to reduce some side effects of cisplatin in cancer therapy [10]. Bi(III) salts (such as BiCl_3_, BiAc_3_, BiO_3_, Bi(n-hexanoate)_3_ and BiSS) have previously been reported to act as catalysts in the copolymerization of CL, Gly, and L-LA in particular [8,11]. These compounds are stable in storage and, most importantly, nontoxic in the quantities needed [8]. Comparing to other nontoxic metal catalysts, bismuth (III) compounds are, therefore, well suited to ROP of lactides [4] and may lead to unusual, random Cap and LL sequences in the polymeric chain [11].

However, there are few reports of bismuth organometallic compounds being used as catalysts of the ROP of lactones. Kricheldorf and Serra [12] published the first reference using BiOct_3_. They highlighted its high effectivity and low tendency of racemization of lactides, even in high temperatures (180 °C) [12]. Though the synthesis of polyesters in the presence of BiOct_3_ has already been published, the purpose of this study was to extend the prior research of Kricheldorf and Serra [12] towards the synthesis of the copolymers of cyclic carboxylic esters, with a potential use as drug carriers in oncology, as well as to evaluate the resulting polymers in relation to their thermal properties and microstructure. The microstructure of polymers influences the kinetics of biodegradation process [13] and is therefore important regarding drug release and, thus, pharmaceutical application.

In this paper, we describe the synthesis of CL, Gly, and L-LA homo- and copolymers in the presence of a biosafe bismuth(III) catalyst system. The structural, physicochemical, and biological properties of these biodegradable polymers were investigated. Low molar mass and dispersity characterize the developed products. Furthermore, they have a random distribution of comonomer units along the polymer chain, resulting in the polyesters being amorphous or having a low degree of crystallinity. Most importantly, the polymers formed are non-toxic. We hope that the polyesters produced can be used in DDSs technology.

## 2. Materials and Methods

### 2.1. Materials

l-Lactide (L-LA, (3S)-*cis*-3,6-Dimethyl-1,4-dioxane-2,5-dione, 98%), ε-caprolactone (CL, 2-Oxepanone, 98 %) and poly(ethylene glycol) (PEG200, *M*_n_ = 200 Da) were purchased from Sigma-Aldrich Co. (Poznań, Poland). Glycolide (Gly, 1,4-Dioxane-2,5-dione, 98%) was purchased from TCI Europe N.V. Co. (Zwijndrecht, Belgium) and bismuth 2-ethylhexanoate from Alfa Aesar Co., part of Thermo Fisher Scientific (Kandel, Germany). Methanol (CH_3_OH, analytical pure), chloroform (CHCl_3_, analytical pure), dichloromethane (DCM, CH_2_Cl_2_, analytical pure) and hydrochloric acid (HCl, 35–38%) were obtained from POCH Co. (Gliwice, Poland).

### 2.2. Synthesis of Homo- and Copolymers via ROP

The polymeric materials were formed in bulk by the ROP of CL, Gly, and L-LA in the presence of a PEG200-BiOct_3_ catalytic system. In brief, appropriate amounts of monomers and PEG200 (1 to 5 g in total) were placed in dry glass ampules (the initiator to monomer ratio was constant as 1:100). Under dry argon, the reaction tubes were degassed and the catalytic amounts of BiOct_3_ were charged. The reaction vessels were sealed and placed in a thermostated oil bath under various conditions, i.e., time and temperature. When the reaction was completed, the polymerization products were dissolved in DCM or chloroform and precipitated in a cold methanol solution containing 5% of HCl (twice) and a cold methanol (last precipitation). The procedure was carried out a total of three times. The isolated polymer was dried in a vacuum oven to a constant weight and stored at 4 °C.

### 2.3. Methods

#### 2.3.1. Structural Analysis of Polymers

Analyses of hydrogen nuclear magnetic resonance (^1^H NMR) and carbon-13 nuclear magnetic resonance (^13^C NMR) were carried out on an Agilent 400 MHz spectrometer at room temperature using CDCl_3_ as a solvent. The spectra were collected using 32 scans (^1^H NMR) or 5000 scans (^13^C NMR) with a 1 s acquisition time.

The copolymer microstructure was characterized by means of the parameters calculated from ^1^H NMR and ^13^C NMR spectra according to the equations presented in the literature: the average length of the lactidyl ($l_{\mathrm{LL}}^{e}$), glycolidyl ($l_{\mathrm{GG}}^{e}$) and caproyl ($l_{\mathrm{Cap}}^{e}$) blocks, randomization ratio (*R*), and transesterification of the second mode (*T*_II_) [14,15,16,17].

The monomer conversion (*conv*_i_) was calculated using ^1^H NMR by comparing integrated signals of equivalent protons from the monomer and the polymer, as follows:(1)$$conv_{i}=\frac{I_{i}}{I_{i}+I_{I}}$$
where *I*_i_ and *I*_I_ are the integral intensities of signals from equivalent protons in the monomer and polymer, respectively.

The microstructures of the obtained copolymers were examined using ^1^H NMR for PCLGA and ^13^C NMR for PLACL in the most convenient spectrum ranges, namely the methylene proton region of GG units and the ε-methylene proton region of Cap units (PCLGA), as well as the carbonyl carbon range of Cap and LL units (PLACL). By analogy with the literature [14,15,16,17], spectral lines were assigned to corresponding comonomeric sequences.

^1^H NMR and ^13^C NMR spectra allow for the calculation of $l_{\mathrm{GG}}^{e}$ and $l_{\mathrm{LL}}^{e}$ using the Equation (2) and $l_{\mathrm{Cap}}^{e}$ using the Equation (3) for ^1^H NMR spectrum and the Equation (4) for ^13^C NMR spectrum, as well as the determination of the contribution of sequences formed as a result of a transesterification process [18].
(2)$$l_{\mathrm{XX}}^{e}=\frac{1}{2}\times \frac{\mathrm{XXX}+\mathrm{XXCap}+\mathrm{CapXX}+\mathrm{CapXCap}}{\mathrm{CapXCap}+\frac{1}{2}\left(\mathrm{XXCap}+\mathrm{CapXX}\right)}$$
(3)$$l_{\mathrm{Cap}}^{e}=\frac{\mathrm{CapCap}+\mathrm{XCap}}{\mathrm{XCap}}$$
(4)$$l_{\mathrm{Cap}}^{e}=\frac{\mathrm{XCapX}+\mathrm{CapCapX}+\mathrm{XCapCap}+\mathrm{CapCapCap}}{\mathrm{XCapX}+\frac{1}{2}\left(\mathrm{CapCapX}+\mathrm{XCapCap}\right)}$$
where X represents glycolyl unit $-{\mathrm{OCH}}_{2}\mathrm{CO}-$ (G) or lactyl unit $-\mathrm{OCH}\left({\mathrm{CH}}_{3}\right)\mathrm{CO}-\left(L\right)$, Cap is caproil unit $-O{\left({\mathrm{CH}}_{2}\right)}_{5}\mathrm{CO}-$, and XCap, XXX, XXCap, and so forth are two- and three-element sequences in the polymer chain.

*T*_II_ may cause scission of glycolidyl or lactidyl units in the copolymer chain leading to the formation of characteristic CapGCap or CapLCap sequences. The yield of *T*_II_ is a quantitative determination of the second mode of transesterification process in the copolymer chain, and was calculated according to the Equation (5):(5)$$T_{\mathrm{II}}=\frac{\left[\mathrm{CapXCap}\right]}{{\left[\mathrm{CapXCap}\right]}_{R}}$$
where [CapXCap] is the experimental concentration of CapXCap sequence and the [CapXCap]_R_ is the concentration of CapXCap sequence in a completely random chain.

The [CapXCap]_R_ can be described by the following relation (Equation (6)) when the ratio of [X]/[Cap] is denoted as *k*‘:(6)$${\left[\mathrm{CapXCap}\right]}_{R}=\frac{k{‘}^{3}}{{\left(k‘+1\right)}^{3}}$$

A degree of the randomness of the copolymer chain was calculated from the Equation (7):(7)$$R=\frac{l_{\mathrm{XX}}^{R}}{l_{\mathrm{XX}}^{e}}$$
where $l_{\mathrm{XX}}^{R}$ and $l_{\mathrm{Cap}}^{R}$ represent the average lengths of glycolidyl or lactidyl (Equation (8)) and caproyl blocks (Equation (9)), respectively, in a completely random copolymer chain.
(8)$$l_{\mathrm{XX}}^{R}=\frac{k‘+1}{2k‘}$$
(9)$$l_{\mathrm{Cap}}^{R}=k‘+1$$

#### 2.3.2. Gel Permeation Chromatography

The molar mass (*M*_n_) and molecular mass distribution (*Đ*) were determined by gel permeation chromatography (GPC) on a Viscotek system comprising GPCmax and TDA 305 (triple detection array (TDA): RI, IV, LS) equipped with DVB Jordi gel column(s) (linear, mixed bed) in DCM as an eluent at 30 °C at a flow rate of 1.0 mL min^−1^.

#### 2.3.3. Cyto- and Genotoxicity

To assess the toxicity of polymeric materials, cytotoxicity and genotoxicity tests were carried out. In brief, the cytotoxicity of polymeric matrices was assessed using the neutral red uptake (NRU) test using BALB/c T3T clone A31 mice fibroblast cell line (American Type Culture Collection) in accordance with the International Organization for Standardization (ISO) 10993-5:2009 Annex A guideline [19]. The polymeric extracts for the assay were formed by incubating the sample in 1 mg mL^−1^ DMEM medium with 5% bovine serum for 24 h at 37 °C. Polyethylene film and latex were used as reference materials.

Genotoxicity of the polymeric materials was evaluated according to ISO 13829:2000 guideline [20] by the *Umu*-test with and without metabolic activation using *Salmonella typhimurium* TA3515/psk1002 (Deutsche Sammlung von Mikroorganismen und Zellkulturen GmbH, Braunschweig, Germany). The polymeric samples were incubated in PBS buffer (GIBCO) for 24 h at 37 °C. The 2-aminoanthracene and 4-nitroquinoline N-oxide were used as positive controls.

#### 2.3.4. Thermal Properties

Thermogravimetric analysis (TGA) was performed with a TGA Q500 V20.7 (TA Instruments) under nitrogen flow (60 mL min^−1^). The measurements were carried out at temperatures ranging from 35 to 600 °C, with a heating rate of 10 °C min^−1^ for the samples placed in an open platinum pan. In order to describe the process of thermal decomposition, the temperatures at which the sample lost 5%, 50%, and 95% of mass (*T*_5%_, *T*_50%_, and *T*_95%_ respectively), as well as the final temperature of thermal decomposition (*T*_f_), were presented. Additionally, the temperature of the maximum rate of thermal decomposition (*T*_max_) was determined as the maximum of differential TGA (DTGA) curve. Furthermore, the mass loss of the sample, i.e., the mass loss as a result of evacuation of residual solvents and moisture at the temperature of 150 °C (Δ*m*_150_), and total mass loss of the sample at a temperature of 600 °C (Δ*m*_t_), were calculated. Δ*m*_t_ values were calculated in relation to the masses at 150 °C.

Differential scanning calorimetry (DSC) measurements were performed using a DSC Q200 instrument (TA Instruments) under nitrogen flow in the temperature range from −140 to 250 °C for the sample placed in aluminum pans, applying a heating rate of 10 °C min^−1^.

For the characterization of the melting and the cold crystallization processes, peak temperatures (*T*_m_ and *T*_c_ respectively), onset temperatures (*T*_on_), and melting and crystallization enthalpies (Δ*H*_m_ and Δ*H*_c_, respectively) were determined. Based on the enthalpy values, the crystalline phase content (crystallinity, *X*_c_) was calculated according to the following Equation (10):(10)$$X_{c}=\frac{\Delta H_{m}-\Delta H_{c}}{\sum_{i}\left(W_{i}\times \Delta H_{\mathrm{mi},100\%}\right)}$$
where Δ*H*_m_ is enthalpy of melting, Δ*H*_c_ is enthalpy of cold crystallization and Δ*H*_mi,100%_ is enthalpy of melting for a fully (100%) crystalline homopolymer (literature data). The values of 106 J g^−1^ [21], 136 J g^−1^ [22] and 191 J g^−1^ [23] for Δ*H*_m,100%_ of PLA, PCL and polyglycolide were used respectively. *W*_i_ is the weight fraction of the Cap, LL and GG co-units in copolymers; for homopolymers, *W*_i_ = 1.

The glass transition temperature (*T*_g_) was evaluated as its midpoint based on the first derivative of DSC curve (dDSC). The temperature value of the minimum of the effect generated on the dDSC curve was taken for this purpose [24]. Additionally, in order to estimate *T*_g_ of the copolymers, a simple Fox Equation (11) was used [25].
(11)$$\frac{1}{T_{g}}=\frac{w_{1}}{T_{g1}}+\frac{w_{2}}{T_{g2}}$$
where *T*_g_ is the glass transition of copolymers constructed from components 1 and 2 with weight fractions *w*_1_ and *w*_2_, respectively. *T*_g1_ and *T*_g2_ are the glass transitions of the individual homopolymers 1 and 2, respectively.

## 3. Results and Discussion

### 3.1. Synthesis and Characterization of Polyesters

Four different polymer matrices were synthesized via ROP of CL, Gly and L-LA in the presence of the simple, inexpensive and nontoxic PEG200-BiOct_3_ catalytic system. Bifunctional PEG200 was applied as a co-initiator, resulting in hydroxyl end-capped linear polyesters. The polymerization process was carried out at 110 °C and 130 °C. The molar ratio of the monomers to the catalyst was 100:1. The number average molecular weight (*M*_n_) of synthesized polymers was controlled by the molar ratio of monomer to co-initiator, which was constant and equal to 100:1.

The ^1^H NMR and ^13^C NMR spectra of the synthesized polymers confirmed their structures. The polymers were obtained with a good monomer conversion (almost equilibrium conversion), acceptable yield and moderately narrow dispersities (*Đ* = 1.23–2.59) (Table 1). The theoretical *M*_n_ of the polyesters was determined based on the original monomer and PEG200 content. However, the *M*_n_ values obtained by GPC were found to be different (lower in most cases) from the theoretical ones. The most probable reason is contamination of the sample with moisture, which results in hydrolysis of monomer to hydroxyl byproducts capable of initiating polymerization. As a result, polyesters with lower *M*_n_ fractions (Figure 1) were produced, leading to an increase in *Đ* [26].

Another explanation for the discrepancy in the data might be a transesterification process that occurred during the polymer chain growth. As a consequence, bond cleavage occurs, creating a modification in the distribution of comonomeric units in the polymer chain [17].

Figure 2 depicts the kinetics of the polymerization process. After 24 h, all monomers show complete conversion, verifying the mentioned earlier hypothesis that bismuth derivative catalysts are effective for ROP of cyclic esters. During the polymerization of PLACL, preferential ring opening towards lactide units was observed. L-LA conversion was almost quantitative after 5 h, but CL conversion was substantially slower, reaching 76%, 86%, and 100% after 5 h, 7 h, and 24 h, respectively. A similar trend was observed for homopolymers (PLA vs PCL). PLA reached almost quantitative conversion of L-LA after 7 h, compared to 93% conversion of CL for PCL.

The polymerization conditions were optimized in terms of time, temperature, and catalyst content, and were selected based on monomer conversion, yield, and *M*_n_ agreement with the theoretical value. The optimal polymerization conditions were set at 24 h and 130 °C, with the exception of PLA, which had an optimum temperature of 110 °C.

The polymerization process was then optimized for the lowest catalyst content while not influencing the properties of the formed polymer matrices (conversion degree, *M*_n_, polydispersity index). Polymers were produced with a high yield and a monomer conversion close to 1. The dispersity values ranged between 1.29 and 1.75, of which the lowest value was observed for PLA. The *M*_n_ of the synthesized polymers was sufficiently consistent with the theoretically expected values (Table 2).

TGA and DTGA curves are presented in Figure 3a,b. The characteristic values of temperature, where the samples reach particular decomposition steps of 5%, 50% and 95% (*T*_5%_, *T*_50%_ and *T*_95%_), as well as *T*_f_ and *T*_max_, are summarized in Table 3. Only small amounts of residues were observed at 600 °C, suggesting complete thermal decomposition of polymers into volatile products.

The data show the following sequence of thermal stability PCL ≥ PCLGA > PLACL > PLA. It was found that PLA was the least thermally stable polymer. However, the presence of Cap units in the copolymer chain significantly increased the thermal stability of PLACL, as shown by the shift of temperature of particular decomposition steps of thermal decomposition (compare PLACL and PLA, Table 3). For example, the shift of *T*_max_ value was equal to 15 °C in relation to PLA. To the contrary, only a small effect of GG units on thermal stability of PCLGA in relation to PCL was observed. In such cases, *T*_max_ shifted 3.5 °C down.

The curves display a one-step degradation in the case of PCL, PCLGA and PLACL (Figure 3a,b). This confirms high homogeneity of both co-polymers, like the PCL homopolymer. However, in the case of PLA, a slight shoulder on DTGA curve (with max at ca. 320 °C, Figure 3b) was observed. This suggests the occurrence of an additional step of polymer decomposition. Despite very low dispersity of PLA (*Đ* = 1.26), it can be supposed that in the first step (at low temperature), the fraction of the polymer characterized by the lower *M*_n_ (Figure 1) started to decompose.

Temperatures of glass transition (*T*_g_), cold crystallization (*T*_c_) and melting (*T*_m_), as well as onset temperatures (*T*_on_) of melting and crystallization, enthalpy of melting (Δ*H*_m_), enthalpy of cold crystallization (Δ*H*_c_) and degree of crystallinity (*X*_c_), are listed in Table 4.

The distribution of the crystallites organization is an important factor influencing the processes. As seen in the DSC thermogram, upon heating of the PLA sample (Figure 4a) at temperature above glass transition, two processes are observed, i.e., crystallization (so called cold crystallization; exothermal effect), followed by melting (endothermal effect). The exothermal process may result from the release of energy due to rearranging of molecules into a lower energy configuration. This results in formation of the better organized (crystalline) phase [27,28]. The molten polymer is characterized by a higher energy compared to the crystalline phase. Due to the changes in the polymer energy states taking place at heating, energy is released or absorbed, which can be observed as exothermal or endothermal effects. Similarly, in the case of PCLGA, both crystallization and melting are observed at heating (Figure 4c). In comparison, only the melting endotherm was observed during heating of PCL (Figure 4b), which can be related to the high crystalline phase content determined for this polymer (*X*_c_ = 98%). On the contrary, no effects of crystallization or melting are observed in the case of PLACL. This suggests an amorphous nature of the copolymer.

The single glass transition (Table 4) was observed in all cases, confirming good homogeneity of the obtained polymers. This process is clearly visible in the case of PLA (from 51.0 °C to 56.4 °C), PLACL (from −15.7 °C to −8.8 °C) and PCLGA (from −59.2 to −54.4 °C), and very difficult for detection in the case of PCL (from −63.1 °C to −62.5 °C). Additionally, the values of *T*_g_, calculated from the Equation (11), for both copolymers PLACL (−17.0 °C) and PCLGA (−52.4 °C) are in good agreement with the measurements (−12.3 °C and −56.2 °C, respectively). 

Thermal properties of all synthesized polymers are in close agreement with the literature data: PLA (*X*_c_ = 35% [29], *T*_g_ = 55–65 °C, *T*_m_ = 145–183 °C [30,31]); PCL (*T*_g_ = −60 °C, *T*_m_ = 60 °C [32]; PLACL/50:50 (*T*_g_ = 4 °C, *T*_m_ = 158 °C [33]); PCLGA of low Gly content (*T*_g_ = −60 °C, *T*_m_ = 54 °C [34,35]). It is known that molecular weight, monomer composition, and crystalline and rigid amorphous fractions development strictly affect thermal properties of polymers [30,33,36]. Therefore, some discrepancies in the results are probably due to chemical and structural differences in materials.

The differences between thermal properties of homo- and copolymers might be discussed in terms of the differences in composition and the reactions leading to their syntheses. Therefore, while PLA and PCL polymers are both crystalline (Figure 3a,b), PLACL is amorphous (Figure 4d). The amorphous state of PLACL may be due to transesterification reactions occurring during the synthesis. Bond scissions in comonomeric units of the copolymer chain lead to shortening of LL and Cap block segments. As a result, the transesterification process increases the randomness of comonomer units along the chain and disrupts the crystallization process, hence reducing the size of crystallites. This conclusion corresponds to the results of D’Auria et al. [33], who analyzed random copolymers of LA and CL. The authors concluded that distribution of comonomers along the chain affects thermal behavior of the copolymer. The copolymers with low comonomer content (LA = 0.1 and CL = 0.1) and long sequences of the prevailing comonomer unit (*l*_Cap_ = 12.2 and *l*_LL_ = 11.1 respectively) were crystalline, while the copolymer with LA content (LA = 0.3) was amorphous, and only small amounts of crystalline phase were observed for the copolymers with the compositions LA = 0.7 and LA = 0.5, for which average lengths of comonomer sequences (*l*_LL_ and *l*_Cap_) were in the range of 1.3–3.2.

In the case of the PCLGA, the melting endotherm with two minima in the temperature range of 10–40 °C was observed (Figure 4c). This might be connected to the presence of two types of crystallites differing in size and/or morphology. Accordingly, small, poorly organized crystallites start melting at lower temperature, while the larger crystallites characterized by the better ordered structure melt at higher temperature. Among the others, this might result due to the possible presence of two different blocks in the copolymer chain: Cap units reach domain and GG units reach domain. However, this explanation seems insufficient considering the appearance of the single glass transition and the course of the thermal decomposition process showing rather good homogeneity of the material. Cold crystallization occurring during heating at low temperature might also lead to the formation of less perfect crystallites due to possible reorganization of the amorphous fraction, as well as the improvement of the structure of the crystallites resulting directly after synthesis. This may result in a diversification of the material, in which blocks of different organization will become present. However, the most likely hypothesis (remaining in substantial consent with the above), seems to be that in this temperature range, two processes (endothermal and exothermal) occur simultaneously, and the thermal effects of these processes overlap. Initially, the melting of the less crystallized fraction (endothermal) begins, followed by melt crystallization (exothermal). The new crystallites formed gradually in this way, with a higher degree of organization, melt at a higher temperature, which is still accompanied by an endothermal effect.

### 3.2. Structural Characterization of the Synthesized Polyester Carriers

As previously stated, the type of catalyst used and the transesterification process during synthesis have a strong influence on the microstructure of the polymers formed [1]. It was demonstrated that the structure of polymers may be controlled by modifying the kind of catalyst and the polymerization process parameters. Higher ROP process temperatures, for example, result in more random copolymers as a result of *T*_II_, which induces redistribution of comonomer units in the polymer chain [1].

The structure of homopolymers and the chain microstructure of copolymers were investigated using ^1^H and ^13^C NMR spectroscopy. The characteristic signals were assigned based on the literature [14,16,37,38] and verified the structure of the obtained PCL, PLA, PLACL and PCLGA.

The ^1^H NMR spectrum of the synthesized PCL: 4.22 ppm (-O-(CH_2_)_5_- C(O)-O-C**H_2_**-CH_2_-O-CH_2_-CH_2_-O-), 4.06 ppm (-O-C**H_2_**-CH_2_-CH_2_-CH_2_-CH_2_-C(O)-), 3.69 ppm (-O-(CH_2_)_5_- C(O)-O-CH_2_-C**H_2_**-O-CH_2_-CH_2_-O-), 3.65 ppm (HO-C**H_2_**-CH_2_-CH_2_-CH_2_-CH_2_-C(O)-) + (-O-(CH_2_)_2_-O-C**H_2_**-C**H_2_**-O-(CH_2_)_2_-O-), 2.31 ppm (-O-CH_2_-CH_2_-CH_2_-CH_2_-C**H_2_**-C(O)-), 1.65 ppm (-O-CH_2_-C**H_2_**-CH_2_-C**H_2_**-CH_2_- C(O)-), 1.38 ppm (-OC-CH_2_-CH_2_-C**H_2_**-CH_2_-CH_2_- C(O)-).

The ^1^H NMR spectrum of the synthesized PLA: 5.17 ppm (-O(O)C-(**H**)C(CH_3_)-), 4.36 ppm (-O(O)C-(**H**)C(CH_3_)-OH), 4.28 ppm (-O-CH_2_-C**H_2_**-O(O)C-(CH_2_)_5_-O-), 3.68 ppm (-O-C**H_2_**-CH_2_-O(O)C-(CH_2_)_5_-O-), 3.62 ppm (-O-(CH_2_)_2_-O-C**H_2_**-C**H_2_**-O-(CH_2_)_2_-O-), 1.59 ppm (-O(O)C-(H)C(C**H_3_**)-).

The examination of the copolymers spectra, namely the ^1^H NMR spectra of PCLGA (CL:Gly = 85:15) (Figure 5) and the ^13^C NMR spectra of PLACL (CL:L-LA = 50:50) (Figure 6), allowed us to assign spectral lines to corresponding comonomeric sequences of PCLGA (Table 5) and PLACL (Table 6). Using the Equations (2)–(9) [14,15,16], the distribution of comonomeric units in the polymer chain was determined in accordance with the published data.

The copolymer of low monomer content (Gly = 0.15 and CL = 0.85) characterized longer sequences of prevailing comonomer unit (*l*_Cap_ = 2.82 and *l*_G_ = 1.02) and low crystallinity (*X*_c_ = 15.1 %), while PLACL (L-LA = 0.5 and CL = 0.5), having similar average lengths of comonomer sequences (*l*_L_ and *l*_Cap_) of 2.80 and 1.41 respectively, was amorphous. The quantitative evaluation of their characteristic sequences (CapGCap and CapLCap) clearly demonstrates the effect of *T*_II_, which lead to unit redistribution in the examined copolymer chains (Table 7). High *T*_II_ value and random structure characterize the copolymers (*R* = 1.07 for PLACL and *R* = 1.33 for PCLGA).

### 3.3. Cyto- and Genotoxicity

The polyesters produced in this work are intended for biomedical uses, such as drug carriers in anticancer DDSs. Despite the fact that the drug cargo has substantial cytotoxic activity, the polymeric matrices are intended to be cell and gene neutral. As a result, the four representatives of synthesized polyesters were evaluated for cytotoxicity and genotoxicity (Table 8).

The NRU test was used for the cytotoxicity testing. The quantitative estimation of viable cells in tested cultures was based on their ability to accumulate the dye in their lysosomes. The viability of BALB/c 3T3 cells was not reduced below 70% as compared to the untreated control by any of the tested dilutions. As a result, all examined polymers may be declared nontoxic in the NRU assay.

The *umu*-test was performed to assess the genotoxic potential of the produced polymeric materials. The growth of *Salmonella typhimurium* determining the toxicity of tested samples was evaluated by a measurement of optical density. All tested samples were not toxic for the *Salmonella typhimurium* (bacteria growth > 0.5) with and without metabolic activation. Furthermore, the induction ratio (*IR*), which represents a sample’s genotoxic potential, was <1.5 for all examined materials. This indicates that none of the produced polymers were genotoxic.

### 3.4. The Possibility of Employing the Produced Polymers as Carriers of Therapeutic Drugs

Polyesters are one of the most significant groups of biodegradable polymers. Homo-, co-, and terpolymers of L-LA, *rac*-LA, CL, and Gly are widely employed in medicine, e.g., as drug carriers [3,6,39]. The advantage of DDSs over traditional drug forms is from controlled and sustained drug release in the body, which is influenced, among other things, by the microstructure of the chain [40] and the composition of the polymer carrier [41]. Polyesters are distinguished mostly by their tunable microstructure and chemistry. As a consequence, their properties may be effectively optimized (e.g., drug release profile, degradation time, structure targeting). In our study, the synthesis conditions were optimized in terms of time, temperature, and catalyst content, which is important for biomedical applications because residuals of the catalyst may contaminate the obtained material. ^1^H and ^13^C NMR studies confirmed the polymer structures and complete monomer conversion. Low molar mass homopolymers and atactic copolymers synthesized in the presence of BiOct_3_ exhibit no cyto- or genotoxicity. Based on our previous experience, the obtained polyesters with a statistical microstructure can be used in the technology of short-term DDSs systems. Furthermore, a high randomization ratio of the polyester chains may be favorable owing to a more uniform drug release profile as a result of the polymer’s homogeneous hydrolytic degradation.

A recent comparison of data from the synthesis of biodegradable polymers using bismuth catalysts has been published [7]. The results reported herein confirm the existing data and demonstrate the tremendous potential of BiOct_3_ to create random polymers. The findings are consistent with Kricheldorf’s research [5,7,42], who has been investigating the utilization of Bi(III) compounds, such as bismuth subsalicylate and bismuth(III) n-hexanoate, in the polymerization of biodegradable polymers. Similarly, in our study the polymers were obtained with a high yield. The catalyst enabled complete monomer conversion with excellent reactivity. BiOct_3_ encouraged random distribution of comonomer units along the chain, and the polymerization process produced polymers with low *Đ*.

We are currently conducting additional research on paclitaxel-DDSs derived from these polymers. The preliminary findings of our research enable us to confirm our hypotheses. In our next article, we will present detailed results from structural, physicochemical, and biological investigations (in vitro and in vivo) of these DDSs.

## 4. Conclusions

Four various biodegradable polymeric matrices were synthesized via ROP of CL, Gly and L-LA in the presence of non-toxic PEG200-BiOct_3_ catalytic system. The catalyst system characterizes high productivity by means of small amounts of the catalyst needed for the polymerization process. The structures of the resulted polyesters correspond well to theoretical assumptions. The polymers showed low polydispersity index and *M*_n_ consistent with theoretical values. The polymers were analyzed by means of their structure and thermal properties. The results are consistent with the literature data. BiOct_3_ catalysts efficiently promoted homo- and copolymerization of CL, Gly, and L-LA in a variable range of monomer compositions, which were coherent with monomer feed ratios. Nevertheless, the transesterification reactions contributed to some extent to the structures with more randomized distribution of monomers along the copolymer chains.

Thermal analysis showed single glass transition temperatures indicating good homogeneity of the polymers. The amorphous nature of PLACL and low ordering of PCLGA deduced from the DSC curves are in accordance with random microstructures of the copolymers.

The results showed that BiOct_3_ is a well-suited catalyst, particularly for L-LA, for which preferential ring opening in relation to CL is observed during the polymerization process. The resulted polymers did not show neither cytotoxicity nor genotoxicity. Additionally, taking into account the extraordinarily low toxicity, BiOct_3_ is a particularly attractive “green” catalyst for ROP of biodegradable polyesters, especially these predicted for contact with the living organisms, including drug delivery systems.

## Figures and Tables

**Figure 1 molecules-27-01139-f001:**
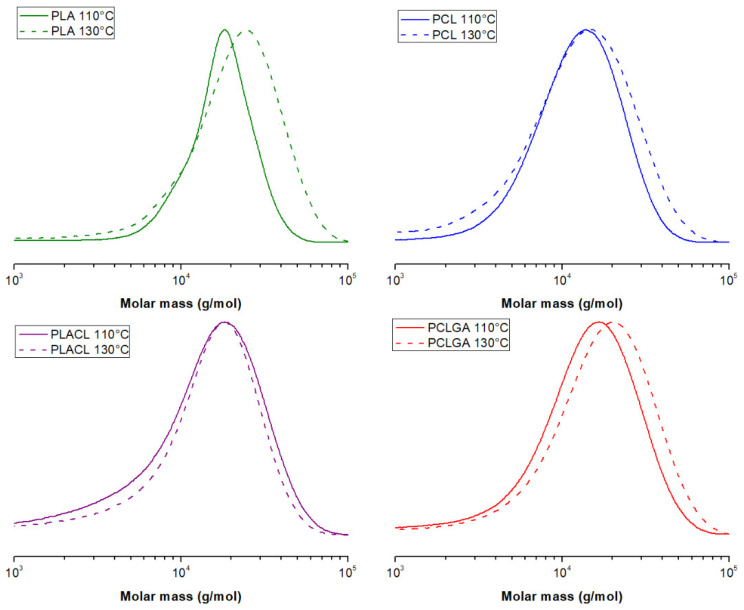
GPC traces for the polymers synthesized in 110 °C (solid line) vs. 130 °C (dashed line).

**Figure 2 molecules-27-01139-f002:**
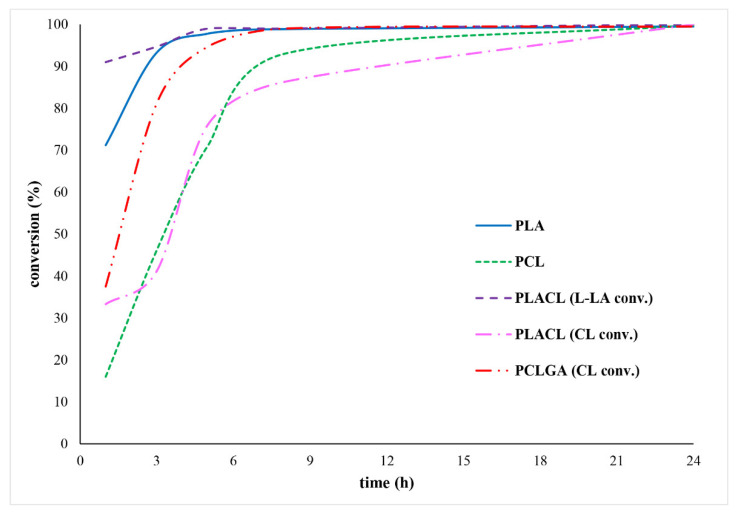
The polymerization kinetics of polymeric carriers (130 °C, PEG200:BiOct_3_ = 1:1).

**Figure 3 molecules-27-01139-f003:**
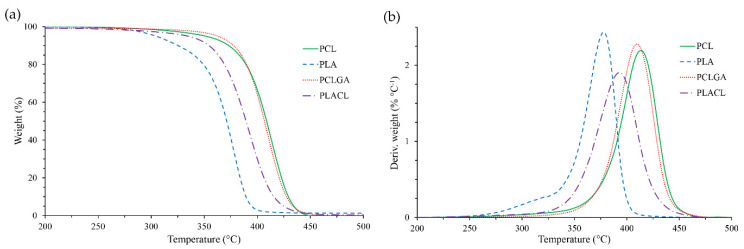
TGA (**a**) and DTGA (**b**) plots of the obtained polymeric carriers.

**Figure 4 molecules-27-01139-f004:**
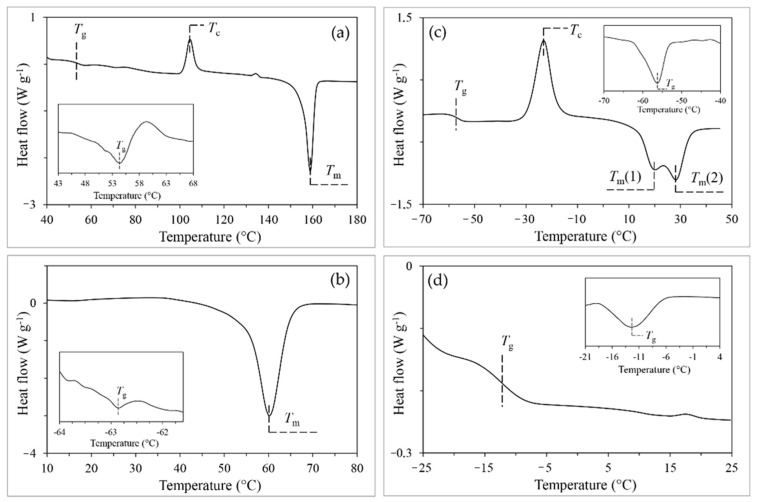
DSC thermograms of (**a**) PLA, (**b**) PCL, (**c**) PCLGA and (**d**) PLACL. DDSC curves in miniatures (determination of *T*_g_).

**Figure 5 molecules-27-01139-f005:**
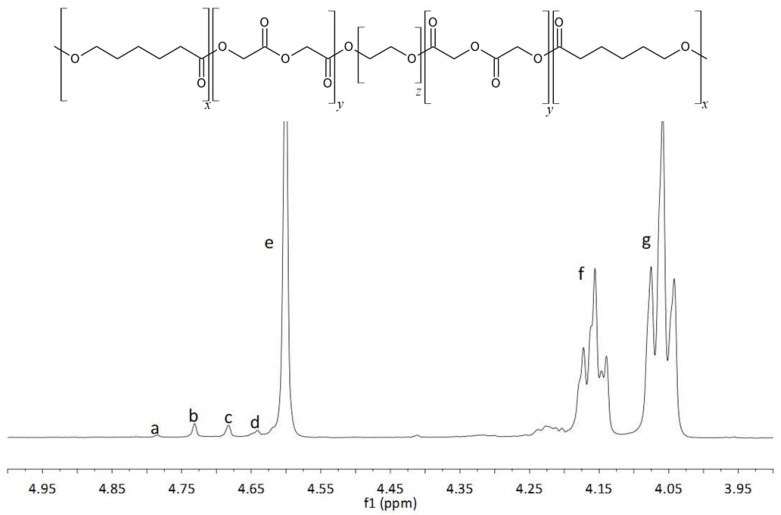
^1^H NMR spectrum of PCLGA.

**Figure 6 molecules-27-01139-f006:**
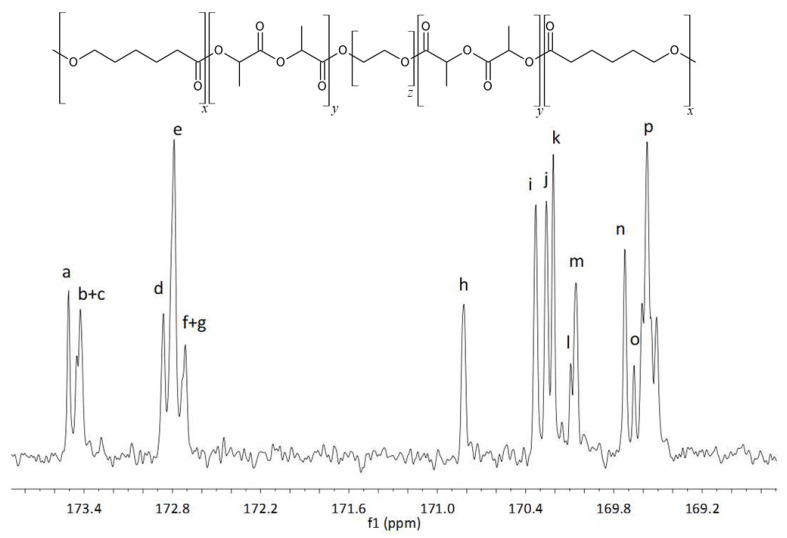
^13^C NMR spectrum of PLACL, region of carbonyl carbon atoms of ε-oxycaproyl and lactidyl units.

**Table 1 molecules-27-01139-t001:** Homo- and copolymerization of CL, Gly and L-LA. Temperature-dependent optimization (24 h, PEG200:BiOct_3_ = 1:1).

Sample	Molar Ratio	Temp. (°C)	Yield (%)	*Conv*_i_^a^ (%)	*M*_n_^b^ (kDa)	*M*_n_^c^ (kDa)	*Đ* ^c^
PLA poly(l-lactide)	L-LA = 1.0	110	73	100	11.7	15.5	1.23
	L-LA = 1.0	130	85	100	13.9	15.3	1.64
PCL poly(ε-caprolactone)	CL = 1.0	110	50	94	11.4	8.8	1.67
	CL = 1.0	130	68	100	12.2	6.9	2.33
PLACL poly(l-lactide-*co*-ε-caprolactone)	L-LA = 0.45 CL = 0.55	110	64	99 (L-LA) 90 (CL)	13.6	7.3	2.50
	L-LA = 0.50 CL = 0.50	130	56	100 (L-LA) 100 (CL)	10.2	9.7	1.87
CLGA poly(ε-caprolactone-*co*-glycolide)	CL = 0.84 GG = 0.16	110	60	100 (Gly) 99 (CL)	11.5	6.8	2.59
	CL = 0.86 GG = 0.14	130	73	100 (Gly) 100 (CL)	10.3	11.8	1.66

^a^—calculated from ^1^H NMR; ^b^—calculated from the feed ratio ^c^—determined from GPC.

**Table 2 molecules-27-01139-t002:** Homo- and copolymerization of CL, Gly and L-LA. Catalyst content-dependent optimization (24 h, 130 °C (110 °C for PLA), monomer:PEG200 = 100:1).

Sample	Monomer/ Catalyst Molar Ratio	Molar Ratio	Yield (%)	*Conv*_i_^a^ (%)	*M*_n_^b^ (kDa)	*M*_n_^c^ (kDa)	*Đ* ^c^
PLA	500	L-LA = 1.0	90	98	12.1	12.3	1.29
PCL	400	CL = 1.0	70	100	11.7	10.8	1.59
PLACL	300	L-LA = 0.52 CL = 0.48	79	99 (L-LA) 99 (CL)	12.6	14.9	1.55
PCLGA	1000	CL = 0.85 GG = 0.15	78	100 (Gly) 98 (CL)	11.8	10.4	1.75

^a^—calculated from ^1^H NMR; ^b^—calculated from the feed ratio; ^c^—determined from GPC.

**Table 3 molecules-27-01139-t003:** Thermal decomposition of polymeric matrices.

Sample	Δ*m*_150_ (%)	Δ*m*_t_ (%)	*T*_5%_ (°C)	*T*_50%_ (°C)	*T*_95%_ (°C)	*T*_max_ (°C)	*T*_f_ (°C)
PLA	0.02	98.77	304.5	371.2	394.6	377.9	414.9
PCL	0.01	100.00	351.4	408.8	434.4	413.3	464.5
PLACL	0.62	99.03	339.3	389.6	424.9	393.3	464.0
PCLGA	0.58	99.50	365.4	406.9	433.8	409.8	475.6

**Table 4 molecules-27-01139-t004:** Thermal parameters of polymeric matrices determined from DSC.

Sample	*T*_g_ (°C)	*T*_c_ (°C)	*T*_on_^a^ (°C)	*T*_m_ (°C)	*T*_on_^b^ (°C)	Δ*H*_c_ (J g^−1^)	Δ*H*_m_ (J g^−1^)	*X*_c_ (%)
PLA	54.4	104.6	102.1	158.9	156.9	15.5	51.4	33.9
PCL	−62.9	nd	nd	60.8	55.7	nd	128.5	98.1
PLACL	−12.3	nd	nd	nd	nd	nd	nd	0.0 ^c^
PCLGA	−56.2	−23.1	−29.1	20.0 28.3	13.8	54.7	76.5	15.1

^a^—cold crystallization process; ^b^—melting process; ^c^—amorphous; nd—not detected.

**Table 5 molecules-27-01139-t005:** Chemical shifts in ^1^H NMR spectrum of PCLGA.

Signal	δ [ppm]	Sequence
a	4.79	G G G
b	4.73	Cap **G** G
c	4.68	G **G** Cap
d	4.64	Cap **G G** Cap
e	4.60	Cap **G** Cap
f	4.16	G **Cap**
g	4.06	Cap **Cap**

**Table 6 molecules-27-01139-t006:** Chemical shifts in ^13^C NMR spectrum of PLACL, region of carbonyl carbon atoms of ε-oxycaproyl and lactidyl units.

Signal	δ [ppm]	Sequence
a	173.59	Cap **Cap** Cap
b	173.45	Cap L **Cap** Cap
c	173.43	L L **Cap** Cap
d	172.86	Cap **Cap** L L
e	172.79	L L **Cap** L L
f	172.73	Cap L **Cap** L Cap
g	172.71	L L **Cap** L Cap
h	170.82	Cap **L** Cap
i	170.33	Cap L L **L** Cap + L L L **L** Cap
j	170.26	Cap L **L** Cap
k	170.21	Cap **L** L Cap
l	170.09	Cap **L** L L Cap
m	170.06	Cap **L** L L L
n	160.73	L L **L** L Cap
o	169.66	Cap L **L** L Cap
p	169.57	L L **L** L L + Cap L **L** L L

**Table 7 molecules-27-01139-t007:** Structural characteristics of PLACL and PCLGA (synthesis parameters: 24 h, 130 °C).

Kind of Copolymer/Molar Ratio	The Average Length of the Blocks	*T* _II_	*R*
poly(l-lactide-*co*-ε-caprolactone) PLACL/50:50	*l*_L_^e^ = 2.80 *l*_CL_^e^ = 1.41	0.70	1.07
poly(ε-caprolactone-*co*-glycolide) PCLGA/85:15	*l*_G_^e^ = 1.02 *l*_CL_^e^ = 2.82	0.96	1.33

*l*_L_^e^—experimental average length of lactyl blocks; *l*_CL_^e^—experimental average length of caproyl blocks; *l*_G_^e^—experimental average length of glycolyl blocks; *R*—randomization ratio; *T*_II_—yield of the second mode of transesterification.

**Table 8 molecules-27-01139-t008:** Results of the *umu*-test and the NRU test in contrast to the untreated control at the highest concentrations of tested extracts [1 mg mL^−1^].

Sample	Genotoxicity Assay		Cytotoxicity Assay
	*IR* ^a^ ± SD	*IR* ^b^ ± SD	Cells Viability ± SD [%]
PLA	0.96 ± 0.02	0.75 ± 0.11	102 ± 2
PCL	0.94 ± 0.11	0.79 ± 0.08	100 ± 1
PLACL	0.87 ± 0.03	0.82 ± 0.14	108 ± 6
PCLGA	1.04 ± 0.11	0.78 ± 0.14	97 ± 4

^a^ version without metabolic activation, ^b^ version with metabolic activation.

## Data Availability

The data presented in this study are available on request from the corresponding author.

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
