# Peer review of "A Comprehensive Investigation of the Structural, Thermal, and Biological Properties of Fully Randomized Biomedical Polyesters Synthesized with a Nontoxic Bismuth(III) Catalyst"

_molecules, 2022, doi:10.3390/molecules27031139_

Round 1

Reviewer 1 Report

The authors used L-lactide, ε-caprolactone and glycolide as monomer to prepared bisdrgdable homopolymers or copolymers, and the synthesized samples were characterized GPC, NMR, TGA, DSC to probe their structure and properties. This manuscript was well written and can be accepted for publication in Molecules after a minor revision.

  • The literatures related with the synthesis and characterization should be cited and discussion. Are there other reports about the synthesis from the monomers to the polymers? If yes, What are the differences in the synthetic methods and the resulting products? What is the novelty in the synthesis and/or the products in this work?

  • More discussion and analyses should be performed on the relation between structure and performance for the synthesized polymers. The authors provided a description about The possibility of employing the produced polymers as carriers of therapeutic drugs. In my opinion, it is not enough to encourage this work to publish in this Journal. It is suggested that the authors correlate more properties with the possible applications. It is better to show a actual application for the synthesized polymer(s).

Author Response

See attatchemnt, please.

Reviewer 2 Report

The manuscript “Synthesis of Fully Randomized Copolymers of ε-Caprolactone, Glycolide and L-Lactide in the Presence of Nontoxic Bismuth(III) Catalyst for Potential Drug Delivery Systems” by  Domańska et al. reports the synthetic methodology of a series of copolymers by Ring Open Polymerization (ROP) employing the PEG200-BiOct3 catalytic system. The work reports the optimal synthetic parameters for the obtention of an aliphatic polyester’s family by variation of several conditions during the synthesis. An exhaustive characterization by NMR and GPC as well as thermoanalytical techniques are included to understand typical aspects of the resulting polymeric structures. Besides, the research briefly covers some studies related to the toxicity of the synthesized polymers demonstrating the fact that could be potentially useful for biomedical applications and highlighting the role of BiOct3 as a green catalyst of ROP polyesters. After a careful reading and evaluation of the submitted manuscript, I consider it an interesting work to be published after some minor changes which are attached below.

  1. I believe the title of the work should be modified due to the authors not demonstrating any potential application in drug delivery. If drug release assays will not be made, please change the title to another more adequate.
  2. Abstract: please, introduce the most important experimental results obtained and some quantitative descriptions.
  3. A brief summary of the work (major details) could be included in the last paragraph of the Introduction.
  4. In Section “2.2. Synthesis of Homo- and Copolymers via ROP” the amounts or ranges should be detailed.
  5. Authors should homogenize the nomenclature to refer to “equations” (please, choose “equations” or “eq” and maintain it along the whole manuscript.
  6. Experimental conditions to carry out the NMR experiments should be explained with major details.
  7. Figures 3 and 4: Please, I recommend the edition of the plots. The thickness of the lines in the graphs could be greater to facilitate the view. Inserts are not visible.
  8. “The exothermal process may result from release of conformational energy due to rearranging of molecules into a ower energy configuration. It result in formation of the better organized (crystalline) phase. Is this supported by similar results in literature? If yes, please, the references have to be added. The same for the following sentence: “The distribution of the crystallites organization is an important factor influencing the processes”.
  9. Regarding structure and NMR results (Figures 5 and 6), I recommend the incorporation of probable chemical structures in order to facilitate the understanding of the peaks and assignations.
  10. Section “3.3. Cyto- and Genotoxicity”: authors should improve the discussion of the presented results.
  11. The first paragraph of section “3.4. The Possibility of Employing the Produced Polymers as Carriers of Therapeutic Drugs”: references are missing.

Author Response

See attatchemnt, please.

Reviewer 3 Report

The manuscript molecules-1586093 is well written and deserves publication in Molecules. Here are some suggestions to improve the presentation of the results.

English should be revised and typos corrected: some examples:

line 87: "are these consisted of metals" should probably be "are those consisting of metals"

line 106: "ultra-element"s should be ultra trace elements.

line 481: "characterize" should be "characterizes"

line 488: "extend" should be "extent"

Author contributions, and statements should be indicated

Structure of the compounds and/or reaction schema should be included.

Author Response

See attatchemnt, please.
